# Quality of Emulsions Based on Modified Watermelon Seed Oil, Stabilized with Orange Fibres

**DOI:** 10.3390/molecules27020513

**Published:** 2022-01-14

**Authors:** Małgorzata Kowalska, Anna Żbikowska, Magdalena Woźniak, Aleksandra Amanowicz

**Affiliations:** 1Department of Management and Product Quality, Faculty of Chemical Engineering and Commodity Science, Kazimierz Pulaski University of Technology and Humanities, 26-600 Radom, Poland; m.wozniak@uthrad.pl (M.W.); amanowicz.aa@gmail.com (A.A.); 2Department of Food Technology and Assessment, Institute of Food Sciences, Warsaw University of Life Sciences-SGGW, 02-772 Warsaw, Poland; anna_zbikowska@sggw.edu.pl

**Keywords:** emulsion, watermelon seed oil, orange fibre, xanthan gum, stability, Turbiscan

## Abstract

The aim of the study was to evaluate emulsion systems prepared on the basis of blended fat in different ratios (watermelon seed oil and mutton tallow) stabilised by orange fibres and xanthan gum. Emulsions were subjected to stability testing by Turbiscan and were assessed in terms of mean droplet size, colour, viscosity, texture, skin hydration and sensory properties. The most stable systems were found to be the ones containing a predominance of mutton tallow in a fat phase. For these emulsions the lowest increase in mean particle size during storage was observed. The study also confirmed the synergistic effect of the thickeners used. The presented emulsions despite favourable physicochemical parameters, did not gain acceptance in sensory evaluation.

## 1. Introduction

Nowadays, research on cosmetics often focuses on the development of innovative products with effective action to meet consumer expectations. When choosing cosmetic products, consumers are usually guided by their function, the components they contain, or they pay attention to the quality, stability and high durability of the product. There is a direct correlation between these characteristics and the cosmetic substrates used in the production process [1].

Emulsion systems represent a large part of the products of the cosmetics industry. They are extremely important both for commercial and scientific reasons and are a frequent object of research. The European Union Directive has introduced restrictions on the origin of raw materials used in the cosmetics industry while preferring ingredients of natural origin or processed mineral raw materials [2]. The use of vegetable oils in cosmetic emulsions is very popular due to their positive effects on the skin resulting from the content of a large amount of unsaturated fatty acids. Moreover, oils are characterised by soothing, moisturising and nourishing properties. Due to their composition similar to skin lipid components, they strengthen the protective barrier and prevent water loss [3].

Innovations in cosmetics are based on global trends, i.e., adding new bioactive substances to their formulations. These are usually oils, ceramides, vitamins, peptides and protein hydrolysates [4]. Following new trends and consumer preferences, these are usually substances that can be used in anti-ageing formulas, antimicrobial products and even pharmaceutical preparations such as analgesics and blood pressure reducers. Increased consumer awareness is resulting in greater interest in formulations containing natural or naturally derived substances (plant or animal) [5].

The innovative ingredient of cosmetic formulations, used in the study, was watermelon seed oil [6]. Traditionally, in order to extract the oil, the seeds are stripped of their skins and left in the sun to dry. The dried seeds are then subjected to a pressing process. Watermelon seed oil contains in its composition more than 75% of unsaturated fatty acids [7]. The highest share is of linoleic and oleic acids, which undoubtedly affects its nutritional and medicinal values. Apart from its use in the cosmetic industry, it can also be used as an edible oil [7].

In addition to the main substances remaining in excess, i.e., water and oil, stabilising substances are included as well in emulsion systems [8]. They can act as emulsion stabilisers by increasing the viscosity of the continuous phase and by acting as a surfactant to form a film around the emulsion droplets [9]. A commonly used emulsion stabiliser is xanthan gum, which is a polysaccharide produced by bacteria of the species *Xanthomonas campestris* [10]. Xanthan gum molecule is built with a cellulose backbone with a trisaccharide side chain at the C-3 atom [11]. This stabiliser usually significantly increases the viscosity of aqueous solutions at low concentrations [12]. Another stabiliser used in this study is the much less common powdered orange fibres [13]. They are obtained from the pomace of oranges used in juice production. Their main function is to protect the emulsion system from phase separation. Orange fibres are not emulsifiers as they do not contain a hydrophobic moiety capable of having an affinity with fats. Fibres added to emulsion systems have a protective (stabilising) effect if the destabilising factor is temperature or pH change [14]. Chatsisvili et al. [15] reported that incorporation of orange pulp rich in fibres resulted in an improvement of low-in-oil dressing-type o/w emulsion stability against creaming.

The main objective of the presented work was to evaluate emulsion systems prepared on the basis of blended fat in different ratios (watermelon seed oil and mutton tallow) stabilised by the natural substances orange fibres and xanthan gum.

## 2. Results

Analysing the results of the mean particle size, it was observed that after 24 h of their preparation the value for all emulsions remained low and at a similar level, ranging from 1.99 µm to 2.95 µm. After a storage period of 30 days, the mean particle size increased slightly. The highest value of mean particle size was recorded for the emulsion with the highest xanthan gum content (Table 1). The smallest increase in particle size after the indicated storage period was recorded for emulsion samples containing predominantly animal fat. In general, no clear effect of the ratio of fats in the fatty phase on the mean particle size was observed. The smallest particle size was characterised by emulsion XIV after 24 h and after the storage period of 30 days, the lowest value of this parameter was noted for emulsion X. On the other hand, emulsion VI had the largest recorded particle size (Table 1). The analysis also confirmed that the average particle size decreased with decreasing xanthan gum concentration to 0.5% in the system.

It was observed that for emulsions containing variable amounts of both stabilisers (increase in pomegranate fibres, decrease in xantan gum amount) the size of emulsion particles reached a smaller size. This trend was observed for the following emulsions (EI, EII, EVI, EVII, EX, EXI). When the content of stabilizers in emulsions was equal the observed trend in the size of emulsion particles changed direction (average particle size increased).

Table 1 also shows the results of the analysis of the mean particle size according to the fat blend used. It was confirmed that for systems that contained a higher share of mutton tallow (three parts by weight) in the fat blend, the value of the mean particle size practically remained unchanged (difference of 0.07 µm). After the storage period, slight increases in this parameter were also observed for the emulsions containing as a fat phase a blend consisted of equal amounts of mutton tallow and vegetable fat. The particle size varied in time for the emulsion with a fat blend in which watermelon seed oil was predominant. Similar information was given by Wozniak et al. [16] who indicated that the average particle size of the emulsion may depend on the fat blend used as a fat phase, consisting of animal fat and vegetable oil.

Viscosity is an important parameter determining the stability of the system, but also a key parameter in consumer evaluation. The lowest viscosity after 24 h as well as after 30 days was observed for emulsions I–V, i.e., systems in which the fatty phase constituted a fat blend with a predominance of watermelon seed oil in its composition. Taking into account varying amounts of xanthan gum and orange fibres no changes in a strictly defined direction were observed. The viscosity of emulsions VI–XV after 24 h was at a similar level. After the storage period for all mentioned emulsions decrease in viscosity was noted, although a smaller decrease was observed for emulsions XI–XV, for which values ranged from 11,308 to 14,892 mPa × s. Generally, it was observed that a greater amount of mutton tallow in the fat phase of an emulsion resulted in a higher viscosity of the system, although this phenomenon was observed only for the measurement performed after 24 h from emulsions preparation. At the same time point, the highest viscosity was noted for emulsion VI, while the lowest for emulsion II. Whereas, after the storage period the highest value of viscosity was recorded for emulsion XII, and the lowest for emulsion V (Figure 1). Thus, the viscosity parameter ambiguously confirmed changes depending on the introduced variable amount of stabilisers. On the other hand, McClements and Jafari [17] pointed out that variable amounts of stabilisers are key factors affecting the viscosity of emulsions and their long-term stability.

The development of new formulations of various cosmetic products and the associated quality control requires tests (monitoring and observation of emulsions over time) to determine their stability. In this field, ageing tests are proposed, but these are considered to be long-lasting and least objective. The phenomena that can occur in emulsion systems are related to their thermodynamic instability. It is well known that these changes relate to phenomena such as flocculation, coalescence, sedimentation, creaming or Ostwald ripening [18]. In recent years, there has been increasing interest in a reliable and rapid method to study the stability of dispersion systems based on the measurement of the intensity of transmitted and backscattered light. In this paper, the stability of emulsion systems was measured using the Turbiscan. This test allows observation of changes in the intensity of light backscattered by the sample or light passing through the sample (transmission) [19]. In general, it can be stated that the curves of the percentage of light transmitted and backscattered by the sample reflect the state of the emulsion as a function of its height. This means that any changes from the standard measurement signal the onset of physical phenomena usually deepening over time. From a thermodynamic point of view, such changes are reasonable and possible because dispersion systems are generally treated as thermodynamically unstable. The curve of the percentage of backscattered light and its changes for emulsions I–XV is shown in Figure 2. It was observed for these emulsions that the changes of the backscattered light values increased in the lower part of the cell and decreased in the upper part. This record confirms the creaming phenomenon. At the same time, it should be noted that the non-overlapping of the curves is also visible in all figures, which indicates the variation of the particle size during the storage period. Such records confirm changes in the type of coalescence or flocculation process. The slowest changes were recorded for emulsion VIII. In their study, Iyer et al. [20] also indicated that the mean particle size is a key parameter in determining the stability of the emulsion system. A larger increase in the particle size of the emulsion system over time results in an apparent non-overlap of the curves when measured with the Turbiscan test.

One of the assessments of the stability of systems, and thus the selection of emulsion components, is to subject them to external influences such as centrifugal force during the centrifugation process. In this study, only emulsions containing as fat base the mixture containing predominantly (three parts) mutton tallow passed this test.

The produced emulsion systems were also subjected to colorimetric examination 24 h after preparation and after 30 days of storage. The highest change in emulsion colour over time was observed for emulsions I–V, while the lowest for emulsions VI–X. The value of L* decreased after 30 days for all tested samples, which means that during storage the colour of emulsions became darker. Analysing the results of the a* parameter values for all emulsions, it was noticed that the values increased in time. After 24 h the a* parameter was negative, which indicates the share of green colour in the emulsion colour. After 30 days of storage in the case of emulsions I–V, a change in the value of the a* parameter to positive, indicating the share of red colour, was observed. The b* parameter increased in time, similarly to a*, but throughout the storage period, it had a positive value indicating the proportion of yellow colour with varied intensity (Table 2).

The emulsions were subjected to texture evaluation 24 h after preparation and after a 30-day storage period. Analysing the results, it was observed, that emulsions I–V containing a predominance of watermelon seed oil were characterised by the lowest hardness and the highest adhesion of all produced systems both on the first day after manufacture and after 30 days. Emulsions VI–X with a predominance of mutton tallow in the fat phase were characterised by the highest hardness, which further increased significantly after 30 days of storage.

The lowest values of adhesive force were also recorded for these emulsions. Storage of emulsions XI–XV caused decrease of this parameter for each mentioned emulsion. On the other hand, emulsions I–V were characterized by an increase in this parameter after the storage period. Referring to hardness it was observed that for emulsions I–V the value of this parameter decreased in relation to the first measurement, i.e., after 24 h from their preparation However, referring to the variable amounts of xanthan gum and orange fibres used, generally, the hardness changed (decreased with the decrease of xanthan gum), while with the increase in orange fibres content the hardness also increased. A similar trend was observed for emulsions XI–XV. For emulsions with the highest proportion of mutton tallow, these changes were less pronounced, which may indicate that hardness was influenced by the amount of solid fat introduced rather than the amount of thickeners. In general, a clear effect of the ratio of fats in the oil phase of the emulsion on its texture was noted (Table 3).

Analysing the results of skin hydration measurement performed on a suitably prepared area of forearm skin, variable skin behaviour after emulsion application was observed. In the case of emulsions I, II, III, the level of hydration immediately after application decreased, but with time gradually increased and after 120 min reached the highest value for emulsion I—40.5 CU; for emulsion II—39.1 CU; for emulsion III—47.6 CU, respectively. In the case of application of emulsions IV and V, the level of skin hydration increased immediately after application reaching a maximum value after 90 min—43.6 CU and 42.6 CU, respectively.

Emulsions VI and X applied to the skin caused an increase in the level of skin hydration immediately after application, but reached its maximum value after the maximum time of the test of 165 min. This confirms that the application of such preparations may be responsible for a long-lasting moisturising effect. In the case of emulsion VI, an increase of only 0.8 CU was observed in relation to the value before application, which suggests the weakest effect and the least beneficial effect of this preparation on the skin. In the case of emulsion VII, there was a decrease in skin hydration after application. This is the only emulsion tested in which this type of behaviour was observed. Emulsions VIII and IX caused an increase in the level of skin hydration reaching its maximum after 60 and 120 min, respectively.

Analysing the results of the increase in skin hydration level for emulsions XI, XIV, XV it can be observed that it reached the highest value after 90 min. Additionally, the highest increase in moisturising values in relation to the result before application was observed with emulsions XIV and XV—by 22.0 CU and 16.7 CU, respectively, which may prove their beneficial oiling and moisturising properties. In the case of emulsions XII and XIII, the level of skin hydration increased in time reaching the highest value after 165 min and 60 min, respectively (Table 4).

The produced emulsions were subjected to sensory analysis for the following parameters: cushion effect, homogeneity, consistency, adhesion, spreading, stickiness, absorption, greasiness and smoothness.

In this evaluation, the highest average score (4.7 points) was obtained by emulsion XIV. On the other hand, the lowest scores were obtained for such features as homogeneity, spreading, stickiness, absorption and smoothness. All other characteristics obtained the maximum 5-points score.

A similar average score (4.6 points) was obtained by emulsions II, XIII, and XV. Lower scores for these emulsions were given to the following parameters: homogeneity, distribution, greasiness, while all other parameters received the maximum score (5 points). The lowest mean scores were obtained by emulsions VI, VII, VIII, IX, and X, which contained in their composition predominance of mutton tallow. The obtained mean values for these systems were respectively: 3.7; 3.4; 3.6; 3.3; 3.6 points. The worst-rated parameters were consistency, greasiness, absorption, adhesion and spreading. The evaluators noted that the emulsions containing the greatest amount of vegetable oil met their preferences the most, although the two emulsions containing a higher content of animal fat (XIII and XV) were also recognised positively (Figure 3). Therefore, it can be assumed that it was not only the type of fat, but also the properties of the thickeners that improved the properties that influenced the final assessment of the respondents.

## 3. Materials and Methods

### 3.1. Materials

Following materials were used: cold-pressed watermelon (WO) seed oil (Ol’Vita, Mysłaków, Poland); mutton tallow (MT) (donated by Meat-Farm Radosław Łuczak, Stefanowo, Poland); sunflower lecithin (L) (Lasenor, Barcelona, Spain); orange fibres as stabiliser (OF) (INCI: Citrus Aurantium Sinensis (Orange) Fiber, Polkowice, Poland); Xantan gum (XG) (BASF, Ludwigshafen, Germany), aqueous solution of sodium benzoate and potassium sorbate (BS) (Schülke & Mayr GmbH, Norderstedt, Germany).

### 3.2. Methods

#### 3.2.1. Emulsion Preparation

Emulsions were prepared according to recipes based on experience [16,21,22]. The oil phase consisted of watermelon seed oil and mutton tallow mixed in the following ratios: 1:3, 1:1 and 3:1. Lecithin was added to the oil phase as an emulsifier. Xanthan gum and orange fibres were dispersed in water (W) using a mechanical homogeniser at 18,500 rpm for 1 min (Table 5). Both phases were heated to 60 °C. Homogenisation of the oil and water phases was carried out using an ultrasonic homogeniser at 30 kHz and 100% amplitude for 2 min. After homogenisation, the emulsions were cooled to room temperature. The preservative (BS) was added for microbiological protection. The total mass of each emulsion was 100 g.

#### 3.2.2. Determination of Destabilisation Changes in Emulsions Using the Turbiscan

A Turbiscan (Formulaction, L’Union, France) was used to observe changes in the emulsions studied. This device uses the principle of multiple light scattering, its back scattering and transmission. This method not only allows the identification and monitoring of instabilities occurring in emulsion systems but also allows their numerical visualisation. It also allows rapid assessment of the stability of emulsion systems without the need for additional sample dilution.

Emulsions I–XV were placed in cylindrical measuring cells with a height of 4 cm and a diameter of 1.6 cm. Samples prepared were stored at 25 °C. During the measurements, the sample was scanned in its entire height. The apparatus recorded the intensity of light transmitted and backscattered by a sample. The light source in Turbiscan is a diode emitting a light beam from the near-infrared range [23].

Measurements were performed at several day intervals for a period of 30 days. The results were given in the form of graphs showing the intensity of backscattered light and its changes as a function of the sample height. The analysis of the graphs for successive measurements allowed determination of the type and intensity of destabilisation changes occurring in the studied samples.

#### 3.2.3. Centrifuge Stability Test

A centrifuge test was carried out to assess the stability of samples under forced conditions. The samples were subjected to centrifugal force by placing them in a centrifuge and subjecting them to 30 min of centrifugation at 3000 rpm [24]. A Nahita Centrifuge Model 2652 (Auxilab, SL, Beriáin, Navarra, Spain) was used and the test was carried out in three cycles of 10 min each, monitoring the condition of each emulsion after the indicated time. Measurements were performed after 24 h from emulsions formation.

#### 3.2.4. Measurement of the Mean Particle Size of Emulsions

The determination of the mean particle size of emulsions was carried out using a microscopic method. Microscopic analysis allows direct observation of the particles of emulsion systems. A Genetic Pro Trino optical microscope (Delta Optical, Warszawa, Poland) equipped with a digital camera DLT Cam Pro (Delta Optical, Gdansk, Poland) was used to take microphotographs of the emulsions. In order to evaluate changes in the particle size, measurements were performed after 24 h from emulsions formation and after 30 days of storage at 2–7 °C.

#### 3.2.5. Colorimetric Study

A colorimetric study was performed using a CR-400 colorimeter (Konica Minolta Sensing Inc., Osaka, Japan). The colour parameters in the CIELAB scale were determined as L*, a*, b* (L*—lightness—ranges from 0 to 100, a*—from green to red, ranges from 120 to 120, b*—from blue to yellow—ranges from 120 to 120). Emulsions were placed in transparent glass vessels of 150 mL capacity. For the precision of the measurement, the instrument was calibrated before the measurements using a standard plate. The test was carried out after 24 h and after 30 days from the formation of the emulsion systems. The result given was the mean of the three measurements.

The total colour difference (TCD) was determined to observe the possible colour change of the emulsions over time. TCD was calculated according to the formula presented by Pathare et al. [25]:(1)$$\mathrm{TCD}=\sqrt{\Delta L^{{*}^{2}}+\Delta a^{{*}^{2}}+\Delta b^{{*}^{2}}}$$
where:ΔL* = (L*_30 days_ − L*_24 h_);(2)
Δa* = (a*_30 days_ − a*_24 h_);(3)
Δb* = (b*_30 days_ − b*_24 h_).(4)
Subscript “24 h” and “30 days” depicts the colour value of emulsions analysed after 24 h and 30 days from their formation, respectively.

#### 3.2.6. Viscosity Measurement

The apparent viscosity of the emulsions was determined by means of Brookfield DV-III Ultra Rheometer, Model—HA with helipath spindle set (Brookfield Engineering Laboratories, Middleboro, MA, USA), using spindle type T-B at the speed of 15 rpm. The measurements were performed at 25 °C. The results were presented as a mean value of three measurements.

#### 3.2.7. Texture Measurement

Emulsions were evaluated for the following texture parameters: firmness, spreadability and adhesiveness using CT3 Texture Analyzer (Brookfield Eng. Laboratories, Middleboro, MA, USA). The detailed procedure was described in our previous study [21].

#### 3.2.8. Assessment of Skin Hydration

Skin hydration of the stratum corneum was assessed using a Corneometer CM 825 (Courage-Khazaka Electronic GmbH, Köln, Germany) on 10 volunteers. The test was carried out in the same room, under repeatable conditions and measuring procedures. Measurements were performed at room temperature—about 21°C, at a humidity of about 50–60% and in the absence of a direct sunlight emission. The test involved measuring skin hydration before and immediately after applying the tested preparations to the skin, and then 30 min, 60 min, 90 min, 120 min, and 165 min after application. For each time point, the measurement was taken in triplicate, then the result was averaged.

#### 3.2.9. Sensory Evaluation of Emulsions

The study was conducted on 10 volunteers, a 5-point score scale was applied (5—maximum score, 1—minimum score). The following parameters were evaluated: cushion effect, homogeneity, consistency, adhesion, spreading, stickiness, absorption, greasiness and smoothness. A detailed procedure is included in the following publication [22]. The same group of volunteers took part in skin hydration and sensory evaluation studies.

## 4. Conclusions

It was found that both the content of watermelon seed oil and mutton tallow in the emulsion and the type of stabilizers used differentiate the properties of the emulsions studied. The most stable were found to be systems VI–X containing a predominance of mutton tallow. The lowest increase in mean particle size during storage was observed for these emulsions. Additionally, these systems passed the stability centrifuge test. Further confirmation of an appropriately composed system was the slight destabilisation changes observed in the curves representing the intensity of backscattered light. The smallest changes in emulsion colour over time were also observed for these systems. Among the five systems presented, emulsion X was characterised by a long-lasting moisturising effect on the skin.

However, it should be noted that these emulsions, despite favourable physicochemical parameters, did not gain acceptance in sensory evaluation. Therefore, the content of mutton tallow in the oil phase requires further study and development, i.e., finding such an amount that will be satisfactory in sensory evaluation and will be characterised by the accepted physicochemical stability.

The study also confirmed the synergistic effect of the thickeners used. Emulsions that contained both thickeners (xanthan gum and orange fibre) in their composition were characterised by more favourable parameters in comparison to emulsions where only one viscosity modifier was used in the formulation.

## Figures and Tables

**Figure 1 molecules-27-00513-f001:**
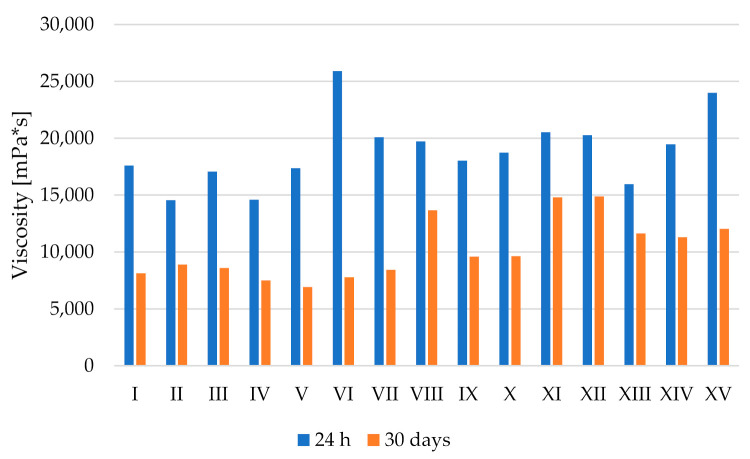
Viscosity of emulsion systems after 24 h and 30 days. Legendary (I–XV) the same as in Table 1.

**Figure 2 molecules-27-00513-f002:**
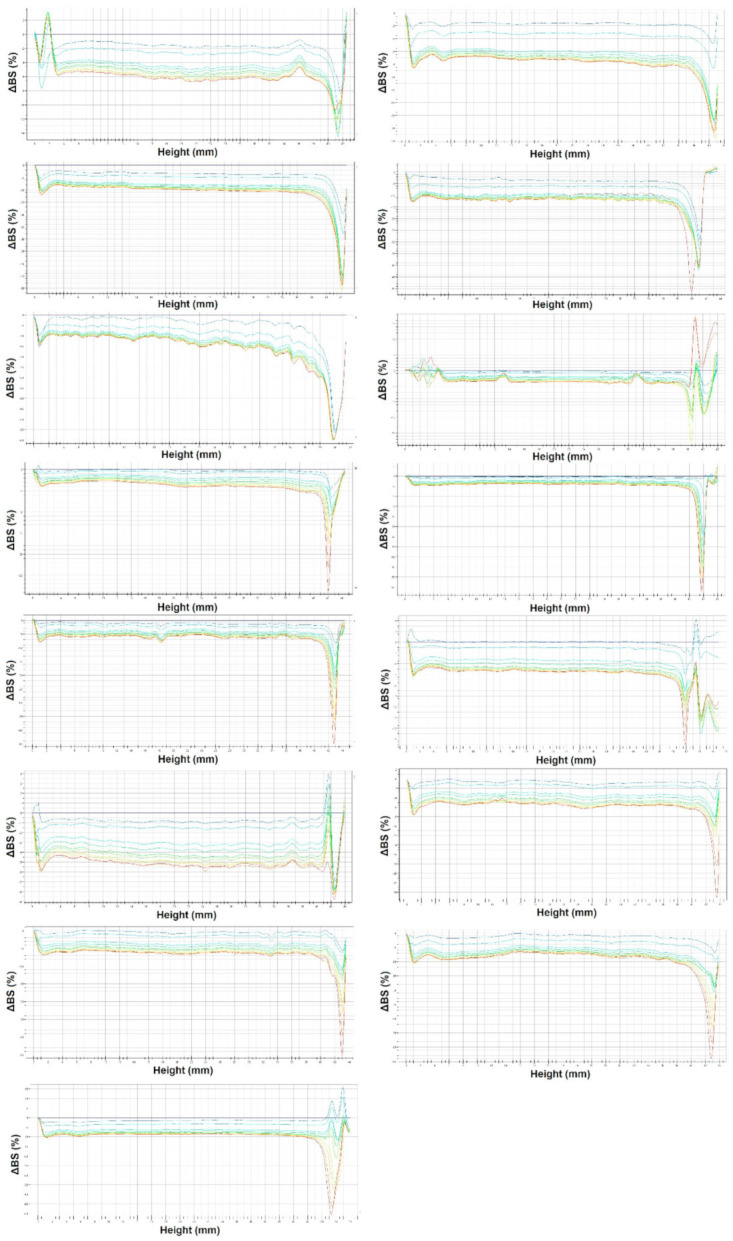
Change in intensity of back-scattered light (ΔBS) over time for emulsions I–XV. Blue line—first measurement, red line—last measurement.

**Figure 3 molecules-27-00513-f003:**
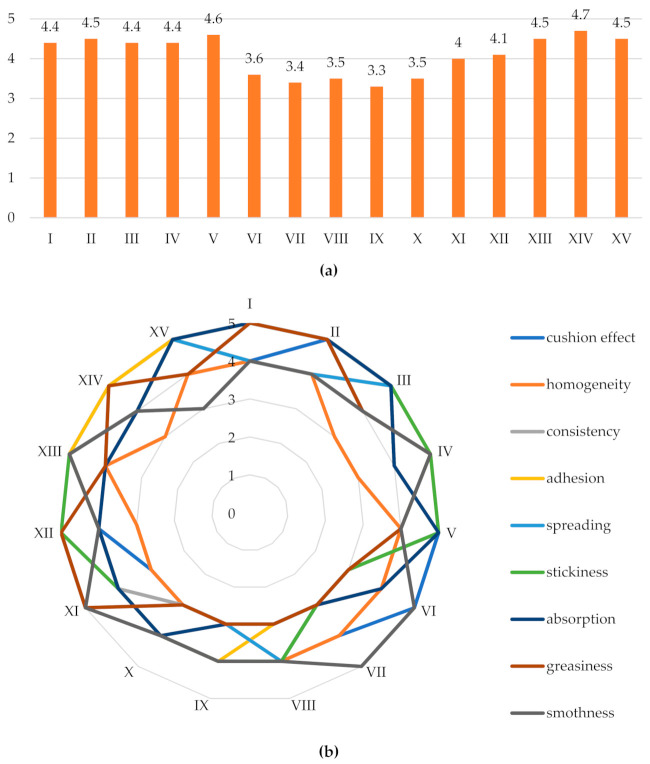
Results of sensory evaluation of the emulsions (**a**) mean scores and (**b**) sensory profile. “1–5“ is a 5-point score scale.

**Table 1 molecules-27-00513-t001:** Particle size of emulsions.

Emulsion	Mean Particle Size (µm)		Increase in Mean Particle Size after Storage Period (µm)	Mean Value of the Particle Size Variation According to the Type of Fat Phase Used (µm)
	24 h	30 Days		
I	2.47	3.24	0.77	0.69
II	2.06	2.82	0.75	
III	2.00	2.91	0.91	
IV	2.02	2.67	0.65	
V	2.23	2.59	0.36	
VI	2.95	3.02	0.07	0.07
VII	2.41	2.47	0.06	
VIII	2.27	2.32	0.05	
IX	2.33	2.49	0.16	
X	2.08	2.09	0.01	
XI	2.61	2.63	0.02	-
XII	2.38	2.49	0.11	
XIII	2.44	2.51	0.07	
XIV	1.99	2.35	0.36	
XV	2.10	2.42	0.32	

**Table 2 molecules-27-00513-t002:** Results of colorimetric study of the emulsion after 24 h and 30 days from their preparation.

Emulsion	Time Point						
	24 h			30 Days			
	L*	a*	b*	L*	a*	b*	TCD
I	63.51	−0.59	9.79	56.86	0.10	12.68	7.28
II	65.17	−0.58	9.79	59.68	0.05	12.59	6.19
III	62.54	−0.59	10.04	58.05	0.17	13.11	5.49
IV	64.78	−0.61	10.07	61.40	0.11	12.42	4.17
V	63.80	−0.62	9.77	59.03	0.02	12.43	5.49
VI	64.14	−0.58	9.74	61.42	−0.22	9.60	2.74
VII	64.55	−0.57	9.61	61.80	−0.22	9.54	2.77
VIII	65.00	−0.64	8.89	62.09	−0.33	9.39	2.96
IX	64.11	−0.64	9.16	61.72	−0.43	9.37	2.41
X	65.85	−0.69	8.47	61.98	−0.30	9.28	3.97
XI	65.11	−0.59	9.67	61.19	−0.17	10.68	4.06
XII	63.26	−0.50	9.82	60.43	−0.23	10.34	2.89
XIII	62.83	−0.56	9.58	59.11	0.03	10.65	3.91
XIV	65.48	−0.65	9.55	61.79	−0.05	10.59	3.88
XV	64.88	−0.69	9.12	60.61	−0.08	10.82	4.63

Meaning of L*, a*, and b* were provided in Materials and Methods section.

**Table 3 molecules-27-00513-t003:** Emulsion texture at 24 h and 30 days after preparation of the systems.

Emulsion	Adhesive Force (g)		Hardness (g)	
	24 h	30 Days	24 h	30 Days
I	−18.0	−5.5	40.5	33.0
II	−10.5	−5.5	33.5	28.0
III	−11.0	−5.5	32.5	24.5
IV	−10.5	−9.0	33.5	25.0
V	−8.0	−7.0	30.5	26.5
VI	−34.0	−89.0	74.5	254.0
VII	−38.5	−159.5	107.0	221.5
VIII	−20.5	−124.0	55.0	222.5
IX	−34.5	−147.0	45.0	240.0
X	−77.0	−128.0	105.5	239.5
XI	−20.0	−23.0	54.5	56.0
XII	−37.5	−38.5	53.5	55.5
XIII	−33.5	−39.0	46.5	48.5
XIV	−33.5	−38.0	57.5	65.5
XV	−43.0	−48.5	80.0	72.5

**Table 4 molecules-27-00513-t004:** Results of skin hydration before application of emulsions I–XV, immediately after application and consecutively after 30, 60, 90, 120, and 165 min.

Skin Hydration [CU]	Emulsion														
	I	II	III	IV	V	VI	VII	VIII	IX	X	XI	XII	XIII	XIV	XV
Before application	31.6	36.4	32.5	34.3	30.3	30.7	30.1	27.5	25.7	28.7	27.8	28.9	21.2	19.9	18.5
Immediately after application	30.4	33.8	32.2	31.7	33.8	20.4	17.8	24.7	20.6	24.3	22.5	26.7	28.8	28.6	35.8
After 30 min	30.3	31.4	36.3	39.0	37.0	21.1	23.8	29.6	23.2	32.6	27.1	29.5	29.4	31.0	35.1
After 60 min	31.7	33.9	41.9	41.2	37.5	21.8	23.6	36.7	26.4	30.3	26.4	21.4	31.9	38.8	35.4
After 90 min	31.3	35.4	42.9	43.6	42.6	20.3	24.1	30.9	26.8	26.6	36.8	20.0	26.6	41.9	35.2
After 120 min	40.5	39.1	47.6	37.8	39.7	28.7	26.6	33.9	32.0	32.4	30.8	22.4	26.9	36.1	31.9
After 165 min	36.6	33.8	36.9	33.5	33.2	31.5	28.7	35.0	29.9	38.0	28.3	34.0	23.2	29.6	27.0

**Table 5 molecules-27-00513-t005:** Emulsions’ composition.

Components		Emulsion														
		I	II	III	IV	V	VI	VII	VIII	IX	X	XI	XII	XIII	XIV	XV
Variable components (% *w/w*)	XG	1.0	0.8	0.5	0.2	0.0	1.0	0.8	0.5	0.2	0.0	1.0	0.8	0.5	0.2	0.0
	OF	0.0	0.2	0.5	0.8	1.0	0.0	0.2	0.5	0.8	1.0	0.0	0.2	0.5	0.8	1.0
	WO	18.375					6.125					12.25				
	MT	6.125					18.375					12.25				
Constant components (% *w/w*)	W	69.0														
	L	5.5														
	BS	q.s.														

XG—xanthan gum; OF—orange fibres; WO—cold-pressed watermelon seed oil; MT—mutton tallow; W—water; L—lectin; BS—preservative; q.s.—quantum satis.

## Data Availability

The data presented in this study are available on request from the corresponding author.

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
