# Peer review of "Quality of Emulsions Based on Modified Watermelon Seed Oil, Stabilized with Orange Fibres"

_molecules, 2022, doi:10.3390/molecules27020513_

Round 1
Reviewer 1 Report
The aim of the study was to evaluate emulsion systems prepared on the basis of blending watermelon seed oil and mutton tallow, stabilised by orange fibres and xanthan gum. Emulsions were subjected to a variety of physicochemical tests and to sensory evaluation. In my oppinion, the main "nolvelties" are associated with the use of the not so usual watermelon seed oil and orange fibres. In general, I see as a good contribution. However, some improvements are necessary:
1) Lines 58-64 - I think that an additional reference can be included with regard to the orange fibres: Nino T. Chatsisvili, Ioannis Amvrosiadis, Vassilis Kiosseoglou, Physicochemical properties of a dressing-type o/w emulsion as influenced by orange pulp fiber incorporation, LWT - Food Science and Technology, Volume 46, Issue 1, 2012, Pages 335-340. Please, consider the possibility of including this reference;
2) Lines 76-77 - The authors affirm, "The smallest particle size was characterised by emulsion XIV both after 24 h and after the storage period of 30 days". Is it correct? Although the particle size for emulsion XIV was also small after the period of 30 days, it was not the smallest (according to Table 1);
3) Lines 79-82 - Please, revise the following phrases (which seems to be valid for part of the cases only): "When the content of this ingredient decreased and the orange fibre content increased, the particle size of the emulsions was found to exhibit higher values than emulsions containing equal amounts of both stabilisers, regardless of the type of fat blend used."
4) Table 1 and table 3 - Please use dots as decimal separators (not commas);
5) Lines 102-106 - It is not possible to follow when the authors affirm, " It was observed that a greater amount of mutton tallow in the fat phase of an emulsion at the same content of both stabilizers resulted in a higher viscosity of the system, although this phenomenon was observed only for the measurement performed after 24h from emulsions preparation". Did the authors refer to emulsion VIII? Emulsion VIII presented higher viscosity for measurements after 30 days (not 24h);
6) Lines 137-138 - The authors state, "This emulsion also showed the smallest increase in average particle size (Table 1)." Although the increase in average particle size for emulsion VIII was small, it was not the smallest (according to Table 1);
7) Figure 2 - Details in the subplots are not visible. Quality of Figure must be improved;
8) Figures 3 and 4 - I think that the results presented in Figures 3 and 4 should also be presented in tables. The presentation in Figures does not allow the precise analysis of the results for skin hydration and sensory evaluation;
9) Line 212 - The authors affirm, "In this evaluation, the highest average score (4.7 points) was obtained by emulsion V." What about emulsion XIV?
10) Lines 216-217 - Distribution?
11) Lines 223-224 - The authors state, "... although the two emulsions containing a higher content of animal fat (XIII and XV)
were also recognised positively". Again: what about emulsion XIV?
12) Line 238 - The authors affirm that emulsions were prepared according to recipes based on their own experience. I think that some details with regard to the authors' experience in preparing the "recipes" (those relevants to the paper presentation) could be included;
13) Table 4 - Please, include units (i.e., g);
Author Response
Radom, December 31, 2021
Article title:
" QUALITY OF EMULSIONS BASED ON MODIFIED WATERMELON SEED OIL, STABILIZED WITH ORANGE FIBERS ”
by Kowalska M, Zbikowska A., Woźniak M, Amanowicz A.
Manuscript ID molecules-1523736
Dear Editor,
The authors would like to thank the reviewers for all kind comments in the reviews. All reviewers' suggestions were taken into account and the text of the manuscript was corrected. Certainly, this will affect the quality of the article and make it more clear (accessible) to the reader. All corrections were introduced into the manuscript in red color. The detailed answers to reviewers’ queries are placed below.
Reviewer 1
The aim of the study was to evaluate emulsion systems prepared on the basis of blending watermelon seed oil and mutton tallow, stabilised by orange fibres and xanthan gum. Emulsions were subjected to a variety of physicochemical tests and to sensory evaluation. In my oppinion, the main "nolvelties" are associated with the use of the not so usual watermelon seed oil and orange fibres. In general, I see as a good contribution. However, some improvements are necessary:
1) Lines 58-64 - I think that an additional reference can be included with regard to the orange fibres: Nino T. Chatsisvili, Ioannis Amvrosiadis, Vassilis Kiosseoglou, Physicochemical properties of a dressing-type o/w emulsion as influenced by orange pulp fiber incorporation, LWT - Food Science and Technology, Volume 46, Issue 1, 2012, Pages 335-340. Please, consider the possibility of including this reference;
The reference was included.
2) Lines 76-77 - The authors affirm, "The smallest particle size was characterised by emulsion XIV both after 24 h and after the storage period of 30 days". Is it correct? Although the particle size for emulsion XIV was also small after the period of 30 days, it was not the smallest (according to Table 1);
It was not correct; the sentence was corrected.
3) Lines 79-82 - Please, revise the following phrases (which seems to be valid for part of the cases only): "When the content of this ingredient decreased and the orange fibre content increased, the particle size of the emulsions was found to exhibit higher values than emulsions containing equal amounts of both stabilisers, regardless of the type of fat blend used."
The paragraph has been correctes.
4) Table 1 and table 3 - Please use dots as decimal separators (not commas);
Decimal separators were replaced by dots.
5) Lines 102-106 - It is not possible to follow when the authors affirm, " It was observed that a greater amount of mutton tallow in the fat phase of an emulsion at the same content of both stabilizers resulted in a higher viscosity of the system, although this phenomenon was observed only for the measurement performed after 24h from emulsions preparation". Did the authors refer to emulsion VIII? Emulsion VIII presented higher viscosity for measurements after 30 days (not 24h);
The sentence was clarified.
6) Lines 137-138 - The authors state, "This emulsion also showed the smallest increase in average particle size (Table 1)." Although the increase in average particle size for emulsion VIII was small, it was not the smallest (according to Table 1);
The sentence was removed.
7) Figure 2 - Details in the subplots are not visible. Quality of Figure must be improved;
Figure was replaced.
8) Figures 3 and 4 - I think that the results presented in Figures 3 and 4 should also be presented in tables. The presentation in Figures does not allow the precise analysis of the results for skin hydration and sensory evaluation;
Figure 3 was replaced by the Table 4. Whereas in Figure 3a we implemented data labelling.
9) Line 212 - The authors affirm, "In this evaluation, the highest average score (4.7 points) was obtained by emulsion V." What about emulsion XIV?
The sentences were corrected.
10) Lines 216-217 - Distribution?
Distribution has not been determinated.
11) Lines 223-224 - The authors state, "... although the two emulsions containing a higher content of animal fat (XIII and XV)
were also recognised positively". Again: what about emulsion XIV?
Since the sentence above was corrected, emulsion XIV was described there.
12) Line 238 - The authors affirm that emulsions were prepared according to recipes based on their own experience. I think that some details with regard to the authors' experience in preparing the "recipes" (those relevants to the paper presentation) could be included;
The references were included.
13) Table 4 - Please, include units (i.e., g);
Units were provided.
Reviewer 2 Report
Comments and Suggestion to Authors _ Round I
molecules-1523736
Title: Quality of emulsions based on modified watermelon seed oil, stabilized with orange fibers
1.) In the title, the species name of watermelon and orange that were used in this study should be provided. In order to let the readers know which species of these plant were used.
2.) In the materials and methods section “3.1. Materials”, the correct species name of watermelon and orange that were used in this study should be provided.
3.) In the materials and methods sections: “3.2.1. Emulsion Preparation”, “3.2.3. Centrifuge Stability Test” and “3.2.4. Measurement of the Mean Particle Size of Emulsions”, the references for these experiments should be provided.
4.) According to the section “3.2.8. Assessment of Skin Hydration” and “3.2.9. Sensory Evaluation of Emulsions”, the authors used human volunteers in their studies, the ethical approval information is need to be provided in their manuscript.
5.) In the section “3.2.8. Assessment of Skin Hydration”, the protocol or the reference for their experimental design should be provided.
6) The authors used 10 volunteers to assess skin hydration and 10 volunteers for sensory evaluation of emulsions. Are these 10 volunteers the same group for the 2 experiments? This information should be provided in the manuscript for the readers.
7) The authors should discuss on the confident of their results from using 10 volunteers to assess skin hydration and 10 volunteers for sensory evaluation of emulsions comparing with the other previous publish works related to their experiments.
8) The authors should add the additional previous published works that related to the scope of their study to discuss with their results. This will help the readers to understand the progression and impacts of this current study.
9.) Some references are not correct following the format of the journal, the authors should check with the instruction for authors, in order to correct this section.
10) There are some spelling mistakes and grammatical errors found in this manuscript, the authors should check and correct during the revision.
Author Response
Radom, December 31, 2021
Article title:
" QUALITY OF EMULSIONS BASED ON MODIFIED WATERMELON SEED OIL, STABILIZED WITH ORANGE FIBERS ”
by Kowalska M, Zbikowska A., Woźniak M, Amanowicz A.
Manuscript ID molecules-1523736
Dear Editor,
The authors would like to thank the reviewers for all kind comments in the reviews. All reviewers' suggestions were taken into account and the text of the manuscript was corrected. Certainly, this will affect the quality of the article and make it more clear (accessible) to the reader. All corrections were introduced into the manuscript in red color. The detailed answers to reviewers’ queries are placed below.
Reviewer 2
1.) In the title, the species name of watermelon and orange that were used in this study should be provided. In order to let the readers know which species of these plant were used.
These products were commercial products, the details were provided in the manuscript. Due to the Christmas break, unfortunately we are unable to obtain this information from the manufacturers at short notice.
2.) In the materials and methods section “3.1. Materials”, the correct species name of watermelon and orange that were used in this study should be provided.
These are raw materials from private companies. Purchased and selected in accordance with the information given on the website of these companies. We were unable to obtain more information.
3.) In the materials and methods sections: “3.2.1. Emulsion Preparation”, “3.2.3. Centrifuge Stability Test” and “3.2.4. Measurement of the Mean Particle Size of Emulsions”, the references for these experiments should be provided.
Emulsions’ preparation was performed based on our own experience – the references were added. We provided the reference for centrifuge stability test. However, we were unable to provide a reference for microscopic observation – this is a basic standard procedure for evaluation of the particle size of emulsions.
4.) According to the section “3.2.8. Assessment of Skin Hydration” and “3.2.9. Sensory Evaluation of Emulsions”, the authors used human volunteers in their studies, the ethical approval information is need to be provided in their manuscript.
The study does not have the features of a medical experiment and does not require the opinion of the bioethics committee. All used materials were food grade, they are well known, safe and commonly sold worldwide. We have the written consent of those taking part in the study.
5.) In the section “3.2.8. Assessment of Skin Hydration”, the protocol or the reference for their experimental design should be provided.
The study of skin hydration is a basic experiment, all applied conditions were provided in the manuscript, and the choice of the time points of the determination is performed by the person who has planned it.
6) The authors used 10 volunteers to assess skin hydration and 10 volunteers for sensory evaluation of emulsions. Are these 10 volunteers the same group for the 2 experiments? This information should be provided in the manuscript for the readers.
Yes, they were the same volunteers.
7) The authors should discuss on the confident of their results from using 10 volunteers to assess skin hydration and 10 volunteers for sensory evaluation of emulsions comparing with the other previous publish works related to their experiments.
The proposed in a study emulsions are novel, we haven’t perform this kind of studies on silimar products.
8) The authors should add the additional previous published works that related to the scope of their study to discuss with their results. This will help the readers to understand the progression and impacts of this current study.
This is a novel study, previously we were studying the effect of interesterification of emulsions’ fatty phases on emulsion systems.
9.) Some references are not correct following the format of the journal, the authors should check with the instruction for authors, in order to correct this section.
The references were verified.
10) There are some spelling mistakes and grammatical errors found in this manuscript, the authors should check and correct during the revision.
The required review has been made.
Round 2
Reviewer 1 Report
The authors kindly addressed all comments and suggestions raised in the "first round" of revision. The quality of presentation was improved.
In this "second round" of revision, I have detected only one minor mistake:
Line 84: The correct term would be ... "orange fibers" (not "pomegranate fibres").
Author Response
Dear Reviewer,
The suggested change has been introduced into the text.
Reviewer 2 Report
Comment and suggestion to authors_Round 2:
Manuscript ID: molecules-1523736
Titled: "Quality of emulsions based on modified watermelon seed oil, stabilized
with orange fibers"
(1) As the plant species provides the different types and amounts of bioactive compounds which effect on the biological activity and efficacy of cosmetic/phytopharmaceutical products. Even through the plant material is “purchased and selected in accordance with the information given on the website of these companies” as informed by the authors, the correct species name of watermelon and orange that were used in this study should be provided in the materials and methods section. So as to offer the clear view and confident for the readers. In fact, there are several species of oranges and watermelon as well.
(2) According to the previous comments:
“7) The authors should discuss on the confident of their results from using 10 volunteers to assess skin hydration and 10 volunteers for sensory evaluation of emulsions comparing with the other previous publish works related to their experiments.”
And
“8) The authors should add the additional previous published works that related to the scope of their study to discuss with their results. This will help the readers to understand the progression and impacts of this current study.”
The authors should provide their discussion with the other previous publish works related to their experiments to emphasize on the novelty and the impacts of this current study. In order to let the readers clearly understand the background and the important to perform this present research.
(3) The references section should be carefully check following the instruction for authors. In addition, the scientific names in references section should be written in italic.
(4) There are some spelling mistakes and grammatical error found in this manuscript, the author should carefully check the whole manuscript before re-submission.
Author Response
Dear Reviewer,
Thank you for your comments and suggestion. The following answers are listed below.
- As the plant species provides the different types and amounts of bioactive compounds which effect on the biological activity and efficacy of cosmetic/phytopharmaceutical products. Even through the plant material is “purchased and selected in accordance with the information given on the website of these companies” as informed by the authors, the correct species name of watermelon and orange that were used in this study should be provided in the materials and methods section. So as to offer the clear view and confident for the readers. In fact, there are several species of oranges and watermelon as well.
The information has been introdoced into the title and the section Material.
According to the previous comments:
- The authors should discuss on the confident of their results from using 10 volunteers to assess skin hydration and 10 volunteers for sensory evaluation of emulsions comparing with the other previous publish works related to their experiments.”
And
The authors should add the additional previous published works that related to the scope of their study to discuss with their results. This will help the readers to understand the progression and impacts of this current study.”
The authors should provide their discussion with the other previous publish works related to their experiments to emphasize on the novelty and the impacts of this current study. In order to let the readers clearly understand the background and the important to perform this present research.
We discussed our previous results with results presented in this work.
- The references section should be carefully check following the instruction for authors. In addition, the scientific names in references section should be written in italic.
References have been prepared acoording quide for authors
- There are some spelling mistakes and grammatical error found in this manuscript, the author should carefully check the whole manuscript before re-submission.
We are not native speakers of English. But as the reviewer suggested, we checked the language again, also asked for an English translator, and corrected the errors as much as possible.